



# PTR-TOF-MS eddy covariance measurements of isoprene and monoterpene fluxes from an Eastern Amazonian rainforest

Chinmoy Sarkar[1], Alex B. Guenther[1], Jeong-Hoo Park[2], Roger Seco[3, 4], Eliane Alves[5, a], Sarah Batalha[6], Raoni Santana[7], Saewung Kim[1], James Smith[8], Julio Tóta[7], and Oscar Vega[9]

[1]Department of Earth System Science, University of California, Irvine, 92697, California, USA
[2]Climate and Air Quality Research Department, National Institute of Environmental Research (NIER), Incheon, 22689, Republic of Korea
[3]Terrestrial Ecology Section, Department of Biology, University of Copenhagen, Copenhagen, Denmark
[4]Center for Permafrost (CENPERM), Department of Geosciences and Natural Resource Management, University of Copenhagen, Copenhagen, Denmark
[5]Department of Climate and Environment, National Institute for Amazonian Research, Manaus, 69067-375, Amazonas, Brazil
[6]Centro Universitário da Amazônia, Universidade da Amazônia, UNAMA, Santarém, 68010-200, Pará, Brazil
[7]Instituto de Engenharia e Geociências, Universidade Federal do Oeste do Pará, Santarém, 68040-255, Pará, Brazil
[8]Department of Chemistry, University of California, Irvine, 92697, California, USA
[9]Centro de Química e Meio Ambiente, Instituto de Pesquisas Energéticas e Nucleares, São Paulo, 05508-000, Brazil
[a]now at Department of Biogeochemical Processes, Max Planck Institute for Biogeochemistry, Jena, 07745, Germany

**Correspondence:** Chinmoy Sarkar (chinmoysarkar8@gmail.com) and Alex B. Guenther (alex.guenther@uci.edu)

**Abstract.** Biogenic volatile organic compounds (BVOCs) are important components of the atmosphere due to their contribution to atmospheric chemistry and biogeochemical cycles. Tropical forests are the largest source of the dominant BVOC emissions (e. g. isoprene and monoterpenes). In this study, we report isoprene and total monoterpene flux measurements with a proton transfer reaction time-of-flight mass spectrometer (PTR-TOF-MS) using the eddy covariance (EC) method at the Tapajós

National Forest (-2.857$^o$S, -54.959$^o$W), a primary rainforest in eastern Amazonia. Measurements were carried out from 1 – 16 June 2014, during the wet to dry transition season. During the measurement period, the measured daytime (06:00 – 18:00 LT) average isoprene mixing ratios and fluxes were 1.15 $\pm$ 0.60 ppb and 0.55 $\pm$ 0.71 mg C m$^{-2}$ h$^{-1}$, respectively, whereas the measured daytime average total monoterpenes mixing ratios and fluxes were 0.14 $\pm$ 0.10 ppb and 0.20 $\pm$ 0.25 mg C m$^{-2}$ h$^{-1}$, respectively. Midday (10:00 – 14: 00 LT) average isoprene and total monoterpenes mixing ratios were 1.70 $\pm$ 0.49 ppb

and 0.24 $\pm$ 0.05 ppb, respectively whereas midday average isoprene and monoterpene fluxes were 1.24 $\pm$ 0.68 mg C m$^{-2}$ h$^{-1}$ and 0.46 $\pm$ 0.22 mg C m$^{-2}$ h$^{-1}$, respectively. Isoprene and total monoterpene emissions in Tapajós were correlated with ambient temperature and solar radiation. Significant correlation with sensible heat flux, SHF (r$^2$ = 0.77), was also observed. Measured isoprene and monoterpene fluxes were strongly correlated with each other (r$^2$ = 0.93). The MEGAN2.1 model could simulate most of the observed diurnal variations (r$^2$ = 0.7 to 0.8) but declined a little later in the evening for both isoprene and

total monoterpene fluxes. The results also demonstrate the importance of site-specific vegetation emission factors (EFs) for accurately simulating BVOC fluxes in regional and global BVOC emission models.



# 1 Introduction

The Amazon rainforest acts as a large photochemical reactor of atmospheric trace gases and aerosols which is significantly influenced by biogenic volatile organic compounds (BVOCs) emitted from the forest (Andreae et al. , 2002). These BVOCs

undergo atmospheric oxidation processes to produce secondary pollutants such as tropospheric ozone ($O_3$) and secondary organic aerosols (SOA) that have significant impact on air quality and climate (Karl et al. , 2010; Pöschl et al. , 2010). Isoprene ($C_5H_8$) dominates the global BVOC budget (500 Tg y$^{-1}$; Guenther et al. (2006)) and plays an important role in the oxidation capacity of the atmosphere due to its high reactivity with the hydroxyl (OH) radical (Claeys et al. , 2004; Karl et al. , 2010). Global monoterpene ($C_{10}H_{16}$) emissions are primarily dominated by $\alpha$-pinene, $\beta$-pinene, t-$\beta$-ocimene, limonene, myrcene,

sabinene, camphene, 3-carene, $\beta$-phellandrene and terpinolene (Guenther et al. , 2012). Although monoterpene emissions are reported to be smaller than isoprene, they are an important class of BVOCs due to their capacity for higher SOA production (Sakulyanontvittaya et al. , 2008; Hallquist et al. , 2009). Emissions of both isoprene and monoterpenes from plants depend on the environmental conditions such as solar radiation, ambient temperature, relative humidity and $CO_2$ concentrations (Kesselmeier et al. , 1999; Guenther et al. , 2006). However, the magnitude of BVOC emissions from most Amazonian tree

species and their variations and distribution over most of the Amazon basin remains unknown.

Although studies related to BVOCs emissions and chemistry in the Amazon rainforest have been carried out in recent decades (Greenberg et al. , 2004; Harley et al. , 2004; Jardine et al. , 2015; Yáñez-Serrano et al. , 2015; Alves et al. , 2016; Liu et al. , 2016, 2018), the extent to which BVOCs control air quality and regional climate through earth system interactions remains poorly understood. The Model of Emissions of Gases and Aerosols from Nature (MEGAN) estimates BVOC

emission fluxes by taking into account the environmental conditions, land use and vegetation emission factors (EFs), based on simple mechanistic algorithms that represent the major processes controlling BVOC emissions (Guenther et al. , 2006, 2012). MEGAN2.1 estimates that $\sim$ 80% of global terpenoid emissions are from tropical forests. Satellite observations suggest lower isoprene emissions in the tropics (Bauwens et al. , 2016). However, aircraft flux measurements during the TROpical Forest and Fire Emissions Experiment (TROFFEE, Karl et al. (2007)) and the Green Ocean Amazon (GoAmazon 2014/5) campaign

(Gu et al. , 2017) both report isoprene emission rates that are not only higher than satellite top-down estimates but are also 35 to 65% higher than MEGAN2.1 estimates (Karl et al. , 2007; Gu et al. , 2017). Therefore, to reduce the uncertainty of the modelled BVOC flux estimations, it is important to estimate site-specific BVOC emission factors (EFs) based on in-situ BVOC flux measurements. Incorporating these site-specific EFs in MEGAN can improve global BVOCs estimation.

Eddy covariance (EC) VOC flux measurements using the proton transfer reaction time-of-flight mass spectrometry (PTR-

TOF-MS) technique were first introduced by Müller et al. (2010), and have since been reported for several other studies (Ruuskanen et al. , 2011; Park et al. , 2013; Acton et al. , 2016; Schallhart et al. , 2018). The major advantage of using a PTR-TOF-MS over a conventional PTR-QMS (with a quadrupole mass analyzer) for VOC flux measurements is the ability of PTR-TOF-MS to measure all masses simultaneously and also to separate the isobaric species based on their monoisotopic masses, allowing us to characterize more VOC species and thus minimize interfering compounds. The simultaneous mass

measurements of the PTR-TOF-MS enables collection of VOC data with 10 Hz frequency for EC estimation which reduces the





random uncertainty of the EC measurement. During a previous comparison study of BVOC fluxes estimated using both PTR-TOF-MS and PTR-QMS in a mixed oak forest in northern Italy, isoprene fluxes were observed to be identical using both the instruments whereas $\sim 25\%$ higher monoterpene fluxes were estimated by PTR-QMS due to the detection of other interfering ions at the same nominal mass (Acton et al. , 2016). The results revealed the importance of using a PTR-TOF-MS to perform

accurate measurements total monoterpene fluxes using the EC approach.

The terpenoid concentration and flux observations reported in this manuscript were conducted in the Tapajós National Forest, located in the eastern Amazonian rainforest, using a commercially available PTR-TOF-MS during 1-16 June 2014. We have previously summarized the observed monoterpene mixing ratios and fluxes (Batalha et al. , 2018). In this manuscript, we also report isoprene concentrations and fluxes and compare the observations with the BVOC fluxes estimated with the MEGAN

2.1 model. The comparison between the measured and the modeled isoprene and monoterpene fluxes are discussed and the importance of site-specific parameters for BVOC flux estimation is highlighted. In addition, the measured BVOC fluxes were compared with previously reported measurements from the Amazon rainforest.

## 2  Methods

### 2.1  Measurement site

The eddy covariance flux measurements of isoprene and total monoterpenes were performed at the Santarem-Km67-Primary Forest (BR-Sa1) tower site in the Tapajós National Forest (-2.857$^o$ S, -54.959$^o$ W), located near km 67 of the Santarém-Cuiabá highway, $\sim 50$ km south of Santarém (Pará state) in north central Brazil. The Tapajós National Forest contains $\sim 450{,}000$ ha of protected old-growth evergreen forest with a closed canopy (mean tree height $\sim 40$ m). It is comparatively drier than the more extensive wet forests in the Amazon basin (Saleska et al. , 2003; Longo et al. , 2018). The forest experiences a 7-month

wet season with total rainfall of $\sim 1920$ mm y$^{-1}$ (Saleska et al. , 2003) and mid-July till mid-December is considered to be the dry season (da Rocha et al. , 2009). Trees in the Tapajós forest show little impact of stress during the dry season as they can access the deep-soil water (Nepstad et al. , 1994; Saleska et al. , 2003). The canopy phenology also plays an important role in the forest's ecosystem sensitivity to droughts (Longo et al. , 2018). It is predicted that the Amazon will become drier in the future due to climate change and therefore, Tapajós can be considered as a model forest for the future Amazon (Cox et al. ,

2000). Figure 1 shows the location of the flux measurement site.

### 2.2  Instrumentation

The BVOC eddy covariance flux measurements were performed using a commercial high-sensitivity PTR-TOF-MS (model 8000; Ionicon Analytic GmbH, Innsbruck, Austria) from 1-16 June 2014, during the transition period between wet and dry seasons in Tapajós. The instrument enables high mass-resolution (m/$\Delta$m > 4000) measurements with a detection limit of a few

ppt (Lindinger et al. , 1998; Müller et al. , 2010; Sarkar et al. , 2016). The PTR-TOF-MS was operated at a drift tube voltage of 600 V and drift tube pressure of 2 mbar, resulting in an E/N ratio of $\sim 136$ Td.



The instrument was calibrated four times (3, 7, 10 and 13 June 2014) during the measurement period using a gravimetric mixture of a 14-component VOC gas standard (Apel-Reimer Environmental Inc., at $\sim$ 500 ppb; stated accuracy better than 8%) containing methanol, acetaldehyde, acetone, isoprene, methacrolein, methyl ethyl ketone, benzene, toluene, o-xylene,

chlorobenzene, $\alpha$-pinene, 1,2,4-trichlorobenzene and 1,3,5-triisopropylbenzene. Calibrations were performed in the range of 2-10 ppb. In order to establish the instrumental background, VOC-free zero air was generated by passing the ambient air through a catalytic converter (stainless steel tube filled with platinum-coated glass wool) heated at 350$^o$C. The measured ion signals were normalized to the primary ion ($H_3O^+$, m/z = 19) as follows (Sarkar et al. , 2016):

$$ncps = \frac{I(RH^+) \times 10^6}{I(H_3O^+)} \times \frac{2}{P_{drift}} \times \frac{T_{drift}}{298.15} \tag{1}$$

The VOC sensitivities did not show any significant change during the four calibrations performed as the instrumental operating conditions remained constant, which is in agreement to several previous studies (de Gouw and Warneke , 2007). During these four calibrations, average isoprene and total monoterpene sensitivities were 12.8 $\pm$ 0.32 ncps/ppb and 14.7 $\pm$ 0.86 ncps/ppb, respectively. Sensitivity for total monoterpenes were estimated by considering the fragmentation at m/z = 137.132 and 81.070. Therefore, the signal measured at m/z = 137.132 was scaled by 2.5, as the calibrations at the above-mentioned

instrumental settings showed $\sim$ 40% of total monoterpenes were detected at m/z = 137.132. This fragmentation pattern of monoterpenes is in agreement with previously reported studies with similar operating conditions (Tani et al. , 2004; Sarkar et al. , 2016). $\sim$ 42% isoprene signal was detected at m/z 41.009 (isoprene fragment) at 136 Td which was also taken into account while calculating the final isoprene mixing ratios.

The limit of detection (LOD) was determined by passing VOC free zero air through the instrument and estimated as the 2$\sigma$

value of the measured normalized signal (Sarkar et al. , 2016). The total uncertainty for the calibrated VOCs were estimated by considering the accuracy error of the VOC standard, the instrumental precision error and the flow fluctuations of the mass flow controllers (MFCs) during the calibrations. The total uncertainty for isoprene and monoterpenes were estimated to be < 20% for this study.

IoniTof software was used for the data acquisition of the 10 Hz mass spectra and the data processing was performed using the

PTRwid data processing tool (Holzinger , 2015). PTRwid has several unique features such as the accurate mass scale calibration and computation of a "unified mass list", from which a robust attribution of mass peaks is possible. The analysis was done on the SumSpectrum stored with the raw mass spectral data, which minimizes the data processing time significantly. The eddy covariance flux calculations were performed using MATLAB software (Mathworks). Further data analysis and plotting of the data were performed using IGOR Pro (WaveMetrics, Inc.).

The VOC sample inlet was located above the forest canopy (height $\sim$ 65 m) and air was sampled continuously through a 100 m long Teflon tube (OD: 3/8", ID: 1/4"; theoretical residence time of air sample in the tube: $\sim$ 4.8 s) at a flow rate of 40 L min$^{-1}$, maintained by a mass flow controller (MKS Instruments, Inc.). A PTFE membrane particle filter (pore size 2 $\mu$m) was used to protect the inlet line from dust and debris. A 3D-sonic anemometer (Applied Technologies, Inc., Boulder, CO) was also installed near the top of the tower which was collocated with the VOC sample inlet. The sonic anemometer was used



to measure air temperature and wind speed components (u = zonal; v = southern and w = vertical) which was used for the
eddy covariance flux calculation. The 10 Hz 3D-sonic anemometer data was recorded by a data logger (CR3000, Campbell
Scientific Inc., Logan, UT).

During the study period, solar radiation data was not collected due to the failure of the pyranometer sensor at the site and
therefore, reanalysis solar radiation data from the MERAA-2 satellite (Modern-Era Retrospective analysis for Research and
Applications, Version 2; Gelaro et al. (2017)) was used. The MERAA-Land data provides hourly average land-surface data
at a horizontal resolution of $0.5^o$ latitude and $0.667^o$ longitude (Reichle et al. , 2011). The solar radiation data along with
air temperature and sensible heat flux data were downloaded from MERAA-Land in the NetCDF format. This NetCDF data
file was then processed by extracting the data for the coordinates of the measurement site ($-2.857^o$S, $-54.959^o$W) and then
the output was saved in .csv format. Figure 2 shows the timeseries and diel box and whisker profiles for the measured air
temperature and sensible heat flux along with estimated air temperature, solar radiation and sensible heat flux obtained from
MERAA-2. Most of the days during the measurement period were sunny while a couple of days experienced cloud cover (on
11 and 13 June 2014). Daytime (06:00 – 18:00 LT) and midday (10:00 – 14:00 LT) average temperatures for the measurement
period were $301.5 \pm 1.5$ K ($\sim 28.4 \pm 1.4^oC$) and $303.0 \pm 1.0$ K ($\sim 29.9 \pm 1.0^oC$), respectively whereas estimated daytime
and midday average temperatures from MERAA-2 were $299.1 \pm 1.7$ K ($\sim 25.9 \pm 1.7^oC$) and $300.2 \pm 1.1$ K ($\sim 27.1 \pm$
130    $1.1^oC$), respectively. There were several gaps on the measured air temperature data due to sensor failure and therefore both air
temperature and solar radiation data obtained from MERAA-2 were used for the BVOC flux estimation using MEGAN model
(section 2.4). The daytime and midday average solar radiation from MERAA-2 were $359.1 \pm 289.5$ W m$^{-2}$ and $663.9 \pm 114.2$
W m$^{-2}$, respectively.

## 2.3   Eddy covariance (EC) method for BVOC flux calculation

135    BVOC fluxes (F$_{wc}$, mg C m$^{-2}$ h$^{-1}$) were determined for 30-minute periods using the eddy covariance method in which vertical
BVOC fluxes were estimated using the mean covariance of deviations for vertical wind speed and individual BVOC mixing
ratios according to the following equation (Park et al. , 2013),

$$F_{wc} = \frac{\sigma}{N} \sum_{i=1}^{N} (w_i - \bar{w}).(c_i - \bar{c}) = \frac{\sigma}{N} \sum_{i=1}^{N} w_i^{'} c_i^{'} \qquad (2)$$

where $\sigma$ = air density (mol m$^{-3}$), N = number of data points, $(w_i - \bar{w})$ or $w_i^{'}$ = instantaneous deviation of vertical wind speed
140    from its mean, $(c_i - \bar{c})$ or $c_i^{'}$ = instantaneous deviation of BVOC mixing ratios from its mean. As per the procedure described
in Park et al. (2013), EC flux error estimations were carried out by considering the systematic errors due to inlet dampening,
sensor separation, instrument response time, random noise in EC flux and uncertainties associated with concentration deter-
minations. Lag-times between wind and VOC data varied during the experiment because they were stored on two different
computers. The lag-time of each 30-minute sample period was visually inspected and manually corrected.





### 2.4 MEGAN 2.1 model for BVOC flux estimation

The Model of Emissions of Gases and Aerosols from Nature (MEGAN 2.1) was used to estimate BVOC fluxes. MEGAN 2.1 is based on simple mechanistic algorithms and estimates BVOC emissions by considering the main processes that drive BVOC emissions (Guenther et al. , 2012). Following Guenther et al. (2012), the BVOC activity factor ($\gamma_i$) can be expressed as,

$$\gamma_i = C_{CE} LAI \gamma_p \gamma_T \gamma_A \gamma_{SM} \gamma_{CO_2} \tag{3}$$

where, $C_{CE}$ is canopy environment coefficient; $\gamma_p$, $\gamma_T$, $\gamma_A$, $\gamma_{SM}$, $\gamma_{CO_2}$ are emission responses to the leaf area index, light, temperature, leaf age, soil moisture and $CO_2$ inhibition activity, respectively. Solar radiation and air temperature data obtained from MERAA-2 were used as an input to run MEGAN 2.1 for this study. Based on the change in LAI, leaf age was estimated by the model. For this study, we assumed that there was no variation in soil moisture ($\gamma_{SM}$) and $CO_2$ ($\gamma_{CO_2}$) inhibition activity. A detailed description of MEGAN 2.1 model settings can be obtained from Guenther et al. (2012).

## 3 Results and Discussion

### 3.1 BVOC mixing ratios and flux

Figure 3a shows the timeseries of 30-minute averaged mixing ratios and fluxes of isoprene and total monoterpenes for the period of 1-16 June 2014. As can be seen from Figure 3a, measured isoprene and total monoterpene mixing ratio variations followed a similar behavior throughout the measurement period. Daytime isoprene and total monoterpene mixing ratios as high as $\sim 4$ ppb and $\sim 0.5$ ppb, respectively, were observed for several days (2, 4, 13 – 15 June) during the measurement period corresponding to the highest isoprene and monoterpene fluxes of $\sim 3.2$ mg C m$^{-2}$ h$^{-1}$ and $\sim 1.1$ mg C m$^{-2}$ h$^{-1}$, respectively. A few days (e. g. 9 – 11 June) had lower daytime isoprene and total monoterpene mixing ratios (peak values of $\sim 2.8$ and $\sim 0.37$ ppb, respectively) and fluxes (peak values of $\sim 1.7$ and $\sim 0.64$ mg C m$^{-2}$ h$^{-1}$, respectively) as expected (Guenther et al. , 2006) from cloudy conditions that result in lower solar radiation and ambient temperature (refer Figure 2). As mentioned in Section 2.2, solar radiation was not measured at the site during the study period. However, the solar radiation data obtained from MERAA-2 satellite for the study period indicated that there were some cloudy days e. g. 11 and 13 June (refer to Figure 2) during the measurement period. Some days during the measurement period had comparatively warmer conditions that led to the higher isoprene and monoterpene emissions. The measured sensible heat flux and ambient temperature, which are also decreased during cloudy conditions, showed similar day-to-day variations that were correlated with isoprene and monoterpenes. The measurement site is dominated by broadleaf evergreen tropical trees which includes some species that are known to have high isoprene and monoterpene emission potentials that are both light and temperature dependent (Kuhn et al. , 2004; Karl et al. , 2007; Guenther et al. , 2012) which is consistent with the observed light dependent emission behavior. The average daytime (06:00 – 18:00 LT) emissions of isoprene ($0.55 \pm 0.71$ mg C m$^{-2}$ h$^{-1}$) and total monoterpenes ($0.20 \pm 0.25$





mg C m$^{-2}$ h$^{-1}$) at the site are relatively low indicating a low fraction of high emitting trees. The maximum emission rates of

both isoprene and monoterpenes are about 5 times higher than the daytime average emission rates.

Figure 3b depicts the box and whisker plots for isoprene and total monoterpene mixing ratios and respective fluxes to show the diel trends. Both isoprene and monoterpene mixing ratios followed a similar trend with that of temperature (refer 2b), with daytime maxima between 14:00 – 15:00 LT (maximum isoprene and total monoterpenes mixing ratios of $\sim$ 2.93 and $\sim$ 0.76 ppb, respectively). On the other hand, the measured isoprene and total monoterpene fluxes tend to follow the diel

profile of solar radiation obtained from MERAA-2 with the maxima between 12:00 – 14:00 LT. This indicates that these biogenic emissions at the Tapajós site are mostly light-dependent and are also stimulated by the increase in temperature. The light-dependent emissions of monoterpenes from Amazonian tree species have previously been reported (Kuhn et al. , 2004; Karl et al. , 2007; Jardine et al. , 2015). However, non-zero nighttime mixing ratios for isoprene and monoterpenes were consistently observed during the measurement period. The measured average nighttime (18:00 – 06:00 LT) isoprene and total

monoterpenes mixing ratios were 1.14 $\pm$ 0.59 ppb and 0.14 $\pm$ 0.09 ppb, respectively, and could be a result of emissions that continue in late afternoon and early evening after the major sinks (photochemical loss and vertical transport) are diminished due to the decreased surface heating and the photochemical production of oxidants. Non-zero nighttime isoprene mixing ratios have previously been observed at other forested sites in central Amazonia (Yáñez-Serrano et al. , 2015; Alves et al. , 2016). In addition, monoterpenes are often emitted at night from pools in specialized storage structures such as resin ducts and glandular

trichomes (Guenther et al. , 2012). Since the average measured temperature at night during the measurement period was only slightly lower than the daytime average temperature ($\sim$ 29$^o$C during daytime while $\sim$ 28$^o$C during nighttime), any light-independent nighttime monoterpene emissions are expected to continue at only slightly lower rates and can build up due to the lower mixing height which can increase concentrations of any compounds emitted at the surface.

### 3.2   Comparison with previous flux studies from Amazon and other tropical forests

Table 1 compares isoprene and monoterpene fluxes reported in this study with previously reported flux studies conducted in the Amazon rainforest and other tropical forests around the world. It is difficult to compare measurements at these sites given the large diurnal and seasonal variations and the lack of a consistent approach for reporting these measurements. We have reported both daytime average, which we define as the measurements made between 06:00 and 18:00 Local Solar time (LST), and midday average for 10:00 - 14:00 LST. We expect that measurements for most of the studies in Table 1 are representative

of conditions that fall somewhere in between these two periods. The daytime, and even the midday, average isoprene fluxes observed in this study (0.55 $\pm$ 0.71 mg C m$^{-2}$ h$^{-1}$ during daytime and 1.24 $\pm$ 0.68 mg C m$^{-2}$ h$^{-1}$ during midday), are lower than the isoprene emission rates reported for previous studies carried out during the dry season and wet to dry transition periods in other locations in the Amazon rainforest which indicates that there are relatively few high isoprene emitting tree species at Tapajós in comparison to other locations in Amazon. This is in agreement with previous leaf (Harley et al. , 2004),

canopy (Rinne et al. , 2002; Müller et al. , 2008) and landscape scale measurements at the Tapajós forest. Harley et al. (2004) combined a leaf enclosure emission survey with plot level tree inventories at 14 Amazon sites and found that the percentage of isoprene emitters ranged from 20.5 to 41.5% across the Amazon Basin. The 3 sites with the lowest percentages, 20.5 to



21.5%, of isoprene emitters were all near the Tapajós km 67 site. Harley et al. (2004) report that a fourth site (km 83) in the Tapajós forest, $\sim$ 16 km south of the km 67 site, had a somewhat higher percentage (29%) of isoprene emitters. Greenberg et al. (2004) deployed a tethered balloon sampling system in the wet season of the year 2000 to measure isoprene concentrations in the mixed layer above the Tapajós km 83 site. They used a simple box model to estimate isoprene fluxes of 2.2 mg C m$^{-2}$ h$^{-1}$, which is about a factor of 2 times higher than the isoprene flux than that we observed at the km 67 site. Tapajós isoprene fluxes in this study were lower than most other tropical forest studies with the exception of fluxes reported for a tropical forest in Costa Rica during an extreme drought period (Karl et al. , 2004) and tropical forests in Borneo, which have relatively low fraction of isoprene emitting trees (Langford et al. , 2010).

In contrast to isoprene, total monoterpene fluxes observed in this study (daytime average flux was 0.20 $\pm$ 0.25 mg C m$^{-2}$ h$^{-1}$ and 0.46 $\pm$ 0.22 mg C m$^{-2}$ h$^{-1}$, respectively) were comparable or higher than values reported for several previous studies carried out in Amazon. Andreae et al. (2002) and Kuhn et al. (2007) observed similar monoterpenes fluxes in Manaus using relaxed eddy covariance methods coupled with GC-MS/FID, while Helmig et al. (1998) reported similar values based on mixed layer concentration measurements in the Peruvian Amazon. Monoterpene fluxes reported in this study were significantly higher than the observed fluxes in the tropical forests in Costa Rica and Borneo (Karl et al. , 2004; Langford et al. , 2010). The monoterpene flux of 0.18 mg C m$^{-2}$ h$^{-1}$ measured at the Tapajos km 83 site by (Greenberg et al. , 2004) is about a factor of 2 lower in contrast to their measured isoprene which is about a factor of 2 higher. Our results suggest that Tapajós has relatively lower isoprene and higher monoterpene emitting tree species in comparison with other locations in the Amazon rainforest but the values reported for the Tapajos km 83 site (Greenberg et al. , 2004) indicate that this could be highly variable even within the Tapajos National Forest. Our results emphasize the diversity of the isoprene and monoterpene emitting tree species distributions in the Amazon rainforest and the importance of in-situ measurements in different parts of Amazon, and at a multiple sites within a region, to obtain regionally representative EFs that are required for accurate BVOC emission estimates using MEGAN or other biogenic VOC emission models.

## 3.3 Comparison of measured BVOC fluxes with MEGAN2.1

The MEGAN2.1 model (Guenther et al. , 2012) estimates isoprene and monoterpene emissions as the product of a landscape average emission factor, based on the emission capacity of the vegetation types in the landscape, and an emission activity factor based on light, temperature and other environmental conditions. MEGAN2.1 has options for assigning emission factors representative of 16 plant function types (PFTs) or using a global map of emission factor distributions that accounts for variations within those PFTs (Guenther et al. , 2012). The MEGAN2.1 emission factors for the evergreen tropical broadleaf tree PFT that dominates in the Tapajós forest is 7 and 1.3 mg compound m$^{-2}$ h$^{-1}$ for isoprene and total monoterpenes, respectively. These values are assigned to all tropical forests when using the 16 PFT emission factor scheme. In contrast, the MEGAN2.1 emission factor distribution map accounts for the variations within tropical forests on a scale of $\sim$ 1 km. For the $\sim$ 3 km $\times$ $\sim$ 3 km region surrounding the Tapajos km 67 study site, the MEGAN2.1 emission factor map average values are 1.98 and 1.16 mg compound m$^{-2}$ h$^{-1}$ for isoprene and total monoterpenes, respectively. The emission factors estimated from the observed isoprene and monoterpene fluxes, 2.2 and 0.8 mg compound m$^{-2}$ h$^{-1}$ respectively, are $\sim$ 10% higher for isoprene and $\sim$ 32%





lower for total monoterpenes. While the MEGAN2.1 PFT-average emission scheme and the MEGAN2.1 1-km emission factor distribution map have similar global averages, this comparison shows that they can differ by more than a factor of 3 locally and demonstrates the importance of using the emission factor distribution map for local to regional scale studies and comparisons
to field measurements.

Figure 4 shows box and whisker plots comparing the diel profiles of measured and MEGAN-predicted isoprene and total monoterpenes emission activity factors, respectively. The emission activity factors for the observed fluxes were calculated as the ratio of the observed emission to the emission factor and provides a direct comparison to the model estimates of the diurnal behavior. The diel profiles in Figure 4 show a strong resemblance between the MEGAN-predicted, based on light
and temperature, and the measured isoprene and total monoterpenes emission activity factors. A delay in the evening decline in emissions was observed for both isoprene and monoterpenes estimated by MEGAN2.1. The diel profiles for measured isoprene and monoterpenes emission activity factors showed slightly different pattern during midday (isoprene is relatively constant from 11:00 to 14:00 while monoterpenes showed a peak around 11:00 to 12:00) which contrasts from the model predictions.

Figures 5a and 5b show scatter plots of the measured and MEGAN-predicted isoprene and total monoterpene fluxes, respectively. Figures 5a and 5b are color-coded by the solar radiation data obtained from MERAA-2 and the measured temperature, respectively. Significant correlations were found between the measured and MEGAN-predicted isoprene ($r^2 = 0.70$) and total monoterpene fluxes ($r^2 = 0.79$) which indicates that MEGAN accounts for most of the short-term variability at the Tapajós site. The differences in the magnitude of the observed and predicted emissions are due to emission factor variability in the Amazon
tropical forests and potentially other factors. The evening delay in the simulated decrease in isoprene may be responsible for the marginally lower correlation as compared to total monoterpenes. As can be seen from the figures, higher isoprene fluxes were always associated with higher solar radiation whereas higher temperatures correspond to higher monoterpene fluxes. Higher nighttime isoprene fluxes ($\sim 0.5$ mg C m$^{-2}$ h$^{-1}$) were observed on a few occasions during the measurement period, most likely due to the evening accumulation of isoprene within the canopy layer. Figures 5c and 5d show the correlations between
the measured isoprene vs total monoterpene fluxes and between the MEGAN-predicted isoprene vs total monoterpene fluxes, respectively. In both cases, we observed strong correlations ($r^2 = 0.93$ and $0.99$, respectively), which further indicates that the isoprene and monoterpene emissions at Tapajós are driven by both solar radiation and temperature.

Figure 6 depicts correlation plots for measured and MEGAN-predicted isoprene fluxes with measured sensible heat flux. Both measured and modeled isoprene fluxes had good correlations with measured SHF ($r^2 = 0.77$ in both cases). The variations
between the light environment within the canopy and the actual temperature contributes to this higher correlation. Further, the measured isoprene fluxes and sensible heat fluxes are interdependent since they use the same sonic anemometer data (Rinne et al. , 2002). However, it is important to note that the MEGAN-predicted isoprene fluxes have a similar correlation with the measured SHF, even though they were calculated using the MERAA-2 derived solar radiation and temperature data, which are independent of the sonic anemometer data. This shows that MEGAN can predict diurnal variations even when driven by
satellite derived meteorology data that have higher uncertainties than measurements. An even more important requirement for accurate isoprene fluxes is to constrain the vegetation EFs for a specific landscape.





## 4 Conclusions

Isoprene and monoterpene fluxes were measured by PTR-TOF-MS eddy covariance at an eastern Amazon rainforest in the Tapajós National Forest during wet to dry transition period (1 – 16 June 2014). The highest measured daytime isoprene and
total monoterpenes mixing ratios were $\sim$ 4 ppb and $\sim$ 0.5 ppb, respectively. The daytime (06:00 – 18:00 LT) average isoprene mixing ratios and fluxes were $1.15 \pm 0.60$ ppb and $0.55 \pm 0.71$ mg C m$^{-2}$ h$^{-1}$, respectively, whereas the measured daytime average total monoterpenes mixing ratios and fluxes were $0.14 \pm 0.10$ ppb and $0.20 \pm 0.25$ mg C m$^{-2}$ h$^{-1}$, respectively. The emissions of isoprene and monoterpenes were dependent on both solar radiation and ambient temperature at Tapajós. The measured isoprene fluxes were comparatively lower than isoprene fluxes reported from other locations in the Amazon rainforest
but similar to previous measurements at the Tapajós km 67 flux tower. However, monoterpenes fluxes were in agreement with several previous studies in Amazon, suggesting that the Tapajós National Forest has a lower fraction of isoprene emitting tree species and has a higher monoterpene/isoprene ratio.

    Comparison of the measured isoprene and total monoterpene fluxes with MEGAN2.1 model estimates suggests that MEGAN2.1 can explain most of the diurnal variations of BVOCs at Tapajós. However, site-specific EFs based on the in-situ measurements
are required to accurately represent the magnitude of the emissions. The emission factors estimated from the observed isoprene and monoterpene fluxes were $\sim$ 10% higher for isoprene and $\sim$ 32% lower for total monoterpenes in comparison to the MEGAN2.1 1-km emission factor distribution map but were 69% lower for isoprene and 38% lower for monoterpenes compared to the MEGAN2.1 PFT approach. This comparison shows that the MEGAN2.1 PFT emission scheme and the MEGAN2.1 1-km emission factor distribution map can differ by more than a factor of 3 locally, although they have similar
global averages. This demonstrates the importance of using the emission factor distribution map for local to regional scale studies and comparisons to field measurements.



*Data availability.* Data used in this manuscript can be obtained by sending an email to chinmoysarkar8@gmail.com or alex.guenther@uci.edu

*Author contributions.* A. G., J-H. P., R. S., S. K., J. S., J. T. and O. V. conceptualized the study. J-H. P, R. S., E. A., S. B., J. T. and O. V. conducted the field measurements. C. S., A. G., J-H. P, R. S., E. A., S. B. analyzed the data. A. G., S. K., J. S. and J. T. supervised the research and administered the project. C. S. and A. G. conducted model simulations and wrote the original draft. All authors reviewed and edited the manuscript. All authors have given approval to the final version of the manuscript.

*Competing interests.* The authors declare no conflicts of interest.

*Acknowledgements.* The authors acknowledge Núcleo de Apoio à Pesquisa no Pará (NAPPA) em Santarém-Pa/Instituto Nacional de Pesquisas da Amazônia (INPA), Programa de Grande Escala Biosfera Atmosfera na Amazônia (LBA) and Instituto Chico Mendes de Conservação da Biodiversidade (ICMBio) em Santarém-Pa for the support during the field campaign. S. B. acknowledges Coordenação de Aperfeiçoamento de Pessoal de Nível Superior (CAPES) and Fundação Amazônia de Amparo a Estudos e Pesquisas (FAPESPA) for her PhD fellowship. A. G. was supported by National Science Foundation (NSF) Atmospheric Chemistry program award AGS1643042.



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



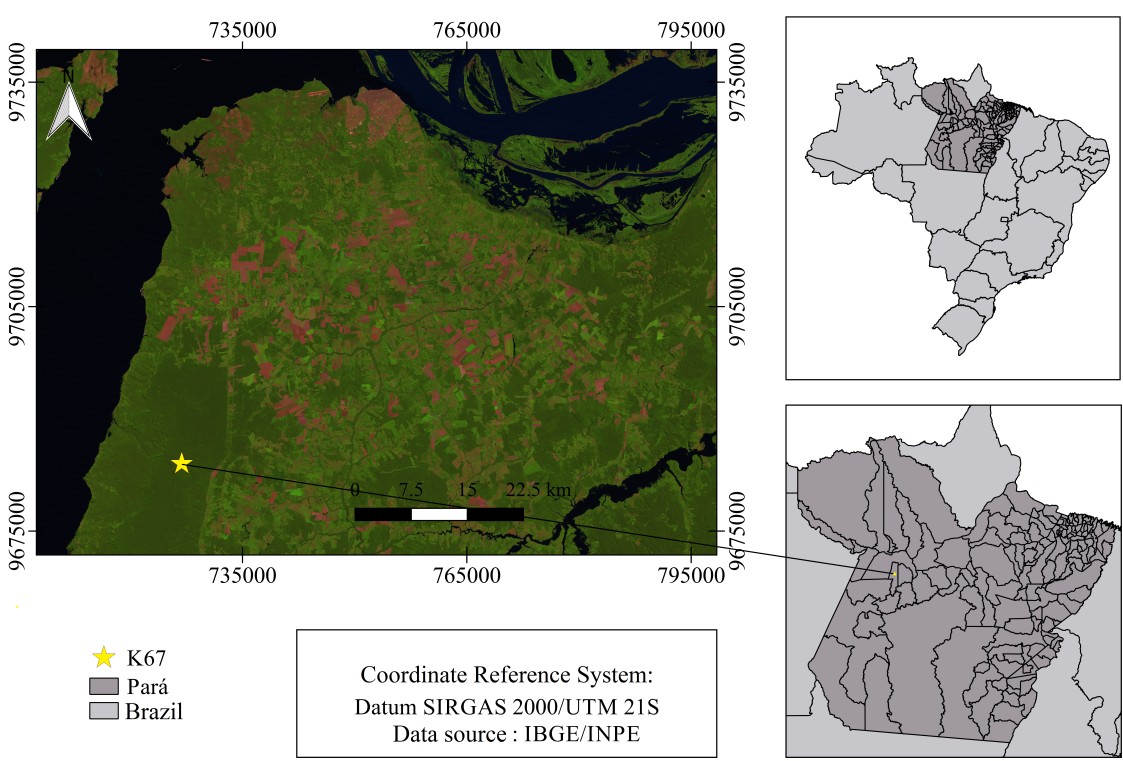

**Figure 1.** Location of the flux measurement site (yellow star) in the state of Pará, Brazil obtained from Landsat imagery courtesy of Brazilian Institute for Space Research (INPE Landsat). The map of Brazil and the state of Pará (right side of the figure) were obtained from the IBGE (Brazilian Institute of Geography and Statistics) website.



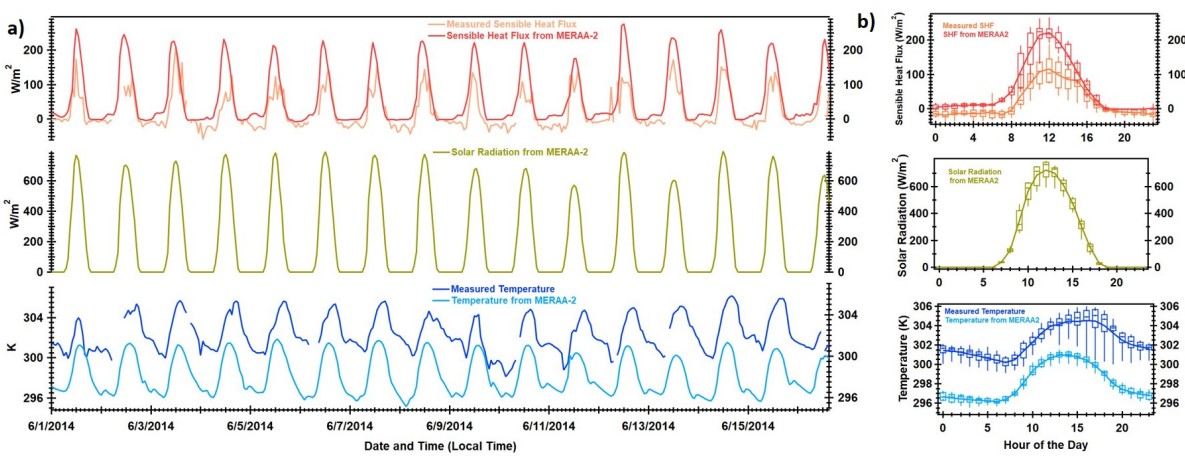

**Figure 2.** a) Timeseries and b) diel box and whisker plots for the measured air temperature and sensible heat flux (SHF) along with estimated air temperature, solar radiation and sensible heat flux obtained from MERAA-2. The timestamp in the X-axis of Figure 2b represents the start-time of the respective hourly data bin (e. g. 6 for data averaged between 6 and 7).





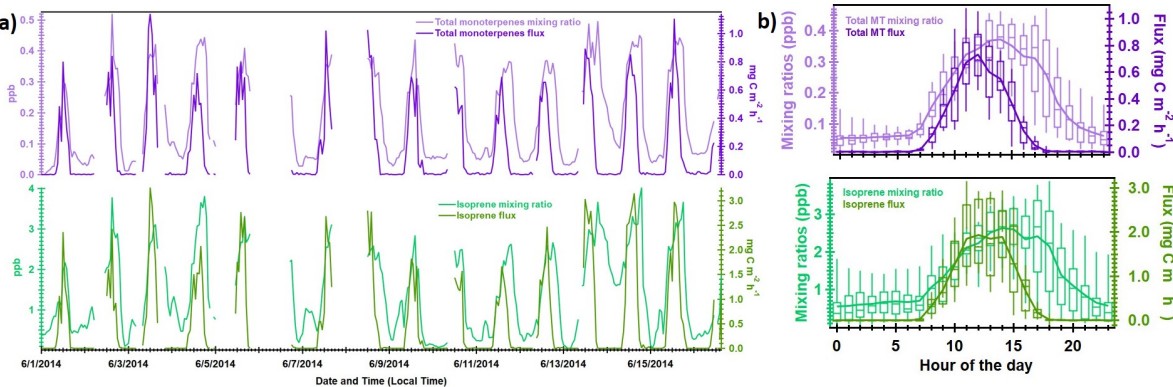

**Figure 3.** a) Timeseries profile for the measured isoprene and total monoterpenes mixing ratios and fluxes, b) box and whisker plots for measured mixing ratios and fluxes of isoprene and total monoterpenes (MT).





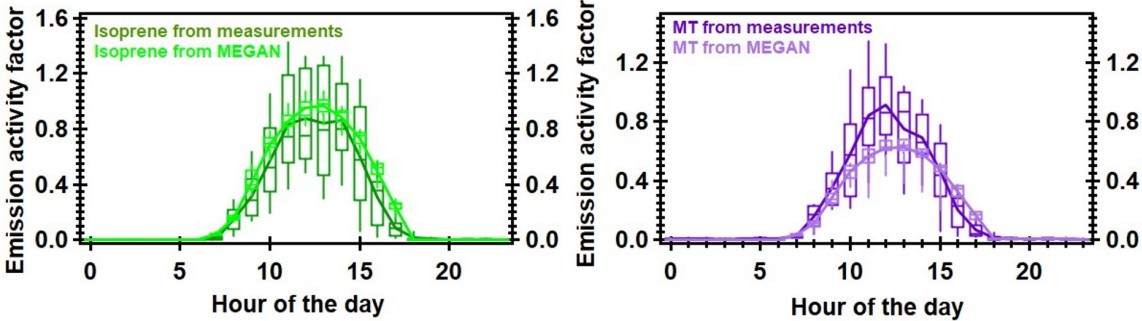

**Figure 4.** Comparison of diel profiles of the emission activity factors calculated based on the measured and MEGAN2.1 predicted isoprene and total monoterpenes fluxes.





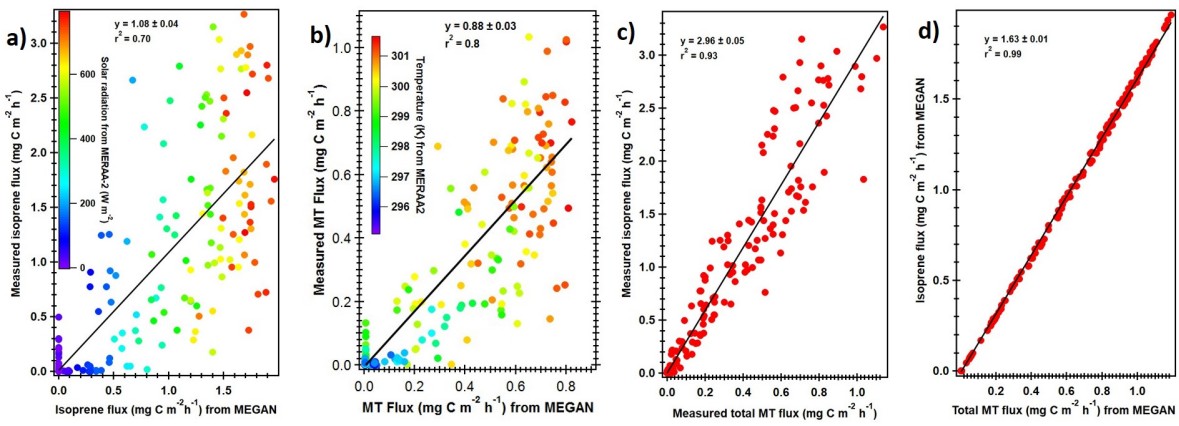

**Figure 5.** Scatter plots for measured and MEGAN2.1 estimated isoprene and total monoterpenes fluxes and their correlations.





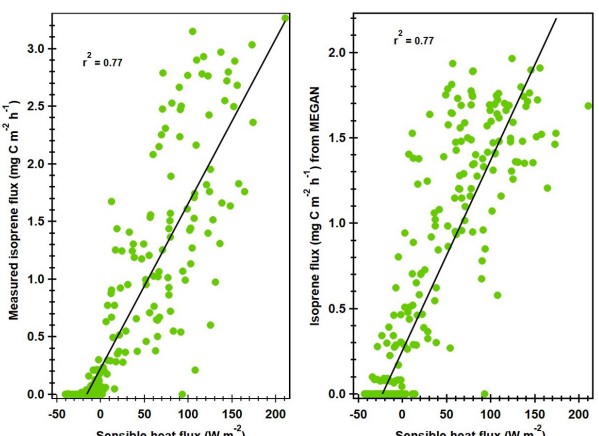

**Figure 6.** Scatter plots for measured and MEGAN2.1 estimated isoprene flux with measured sensible heat flux.





**Table 1.** Comparison of average daytime isoprene and monoterpenes fluxes and their typical ratios ($\sum$ monoterpenes/isoprene) at different locations in the Amazonian rainforest and other tropical forests in the world

| Location | Season | Method | Isoprene (mg C m$^{-2}$ h$^{-1}$) | $\sum$ Monoterpenes (mg C m$^{-2}$ h$^{-1}$) | MT/Iso Ratio | References |
|---|---|---|---|---|---|---|
| Amazon, Peru | Dry (July 1996) | GC-MS, cartridge samples mixed layer gradient | 7.4 | 0.42 | 0.06 | Helmig et al. (1998) |
| Costa Rica | Dry (April – May 2003) | PTR-MS, disjunct eddy covariance | 0.29 | 0.13 | 0.45 | Karl et al. (2004) |
| Borneo, SE Asia | Wet | PTR-MS, disjunct eddy covariance | 0.48 ± 0.72 | 0.13 ± 0.19 | 0.27 | Langford et al. (2010) |
| | Dry | | 1.04 ± 1.3 | 0.25 ± 0.33 | 0.24 | |
| Ndoki, Congo | Wet-to-dry (March 1996) | GC-RGD/MS, Teflon bag, relaxed eddy accumulation | 1.4 ± 1.05 | < 0.1 | < 0.1 | Serça et al. (2001) |
| | Dry-to-wet (November 1996) | | 0.46 ± 0.47 | < 0.05 | < 0.1 | |
| ABLE – Adolfo Ducke-Forest Reserve, Manaus, Brazil | Dry (July – Aug 1985) | GC-FID, Teflon bag on tethered balloon (up to 305 m) | 3.1 | 0.23 | 0.07 | Zimmerman et al. (1988) |
| Balbina, ∼ 100 km north of Manaus, Brazil | Wet (Mar 1998) | GC-MS, mixed layer box model | 5.3 | 0.23 | 0.04 | Greenberg et al. (2004) |
| Jaru Biological Reserve, Jaru-Rondônia, Brazil | Wet (Feb 1999) | GC-MS, mixed layer box model | 9.8 | 6.1 | 0.62 | Greenberg et al. (2004) |
| Cuieiras Biological Reserve (K34-ZF2), Manaus, Brazil | Dry-Wet (Nov 1999 – Jan 2000) | GC-MS, cartridge on relaxed eddy accumulation (∼ 53 m) | 2.88 | 0.36 | 0.13 | Andreae et al. (2002) |
| Cuieiras Biological Reserve (K34-ZF2), Manaus, Brazil | Dry (July 2001) | GC-FID, cartridge, relaxed eddy accumulation (∼ 51 m) | 2.4 ± 1.8 | 0.44 ± 0.49 | 0.18 | Kuhn et al. (2007) |
| | | GC-FID, cartridge, surface layer gradient (28, 35.5, 42.5, 51 m) | 3.9 ± 4.1 | 0.43 ± 0.65 | 0.11 | |
| Cuieiras Biological Reserve (TT34-ZF2), Manaus, Brazil | Dry (Sep – Oct 2010) | PTR-MS, gradient profile (2, 11, 17, 24, 30 and 40 m) and gradient flux | 1.37 ± 0.7 | 1.47 ± 0.06 | 1.07 | Alves et al. (2016) |
| | Dry to Wet transition (Nov 2010) | | 1.41 ± 0.1 | 1.29 ± 0.2 | 0.91 | |
| | Wet (Dec 2010 – Jan 2011) | | 0.52 ± 0.1 | 0.36 ± 0.05 | 0.69 | |
| Cuieiras Biological Reserve (K14-ZF2), Manaus, Brazil | Dry (Sep 2004) | PTR-MS, disjunct eddy covariance (∼ 54 m) | 8.3 ± 3.1 | 1.7 ± 1.3 | 0.21 | Karl et al. (2007) |
| | | PTR-MS, mixed layer gradient (up to ∼ 1200 m) | 12.1 ± 4.0 | 3.5 ± 1.2 | 0.29 | |
| Cuieiras Biological Reserve (K14-ZF2), Manaus, Brazil | Dry (Sep 2004) | PTR-MS, disjunct eddy covariance (∼ 54 m) | 8.4 | 0.93 | 0.11 | Rizzo et al. (2010) |
| Tapajós National Forest (km 83 site) - Pará Santarém, Brazil | Wet (Jan – Feb 2000) | GC-MS, mixed layer box model | 2.2 | 0.18 | 0.08 | Greenberg et al. (2004) |
| Tapajós National Forest (km 67 site) - Pará Santarém, Brazil | Wet to Dry (July 2000) | Disjunct eddy accumulation (∼ 45 m), fast isoprene sensor (FIS) | 1 | 0.1 | 0.1 | Rinne et al. (2002) |
| Tapajós National Forest (km 67 site) - Pará Santarém, Brazil | Wet (April 2001) | Chemiluminescence, eddy covariance-fast isoprene sensor (∼ 70 m) | 0.13 | | | Müller et al. (2008) |
| | Dry (Oct-Nov 2003) | | 1 | | | |
| Tapajós National Forest (km 67 site) - Pará Santarém, Brazil | Wet to Dry (June 2014) | PTR-TOF-MS, eddy covariance (∼ 65 m) | 0.55 ± 0.71 (daytime) | 0.20 ± 0.25 (daytime) | 0.36 | This Study |
| | | | 1.24 ± 0.68 (midday) | 0.46 ± 0.22 (midday) | 0.37 | |