# Peer review of "PTR-TOF-MS eddy covariance measurements of isoprene and monoterpene fluxes from an Eastern Amazonian rainforest"

_Atmospheric Chemistry and Physics, 2019_

## Referee Comment (RC1) · Anonymous Referee #1 · 10 Mar 2020

This study reports BVOC concentrations and fluxes measured at an eastern Amazon rainforest in the Tapajós National Forest during wet to dry transition period, and compare the observations with the BVOC fluxes estimated with the MEGAN 2.1 model. The comparison between the measured and the modeled isoprene and monoterpene fluxes are discussed and the importance of site-specific parameters for BVOC flux estimation is highlighted. In addition, the measured BVOC fluxes were compared with previously reported measurements from the Amazon rainforest. The results demonstrate the importance of site-specific vegetation emission factors for accurately simulating BVOC fluxes in regional and global BVOC emission models. In general, all measured BVOC concentrations and emission fluxes are very valuable for accurate understanding of

BVOCs and their chemistry and photochemistry, BVOC model simulations at regional and global scales. Some minor comments and suggestions are as follows.

Line 100, "The total uncertainty for isoprene and monoterpenes were estimated to be < 20% for this study", Please explain what is the total uncertainty for isoprene and monoterpenes, including the measurements, e.g., sample collection, pump, etc.?

Line 141, what is the EC flux error estimation? Please explain it.

Line151, "Solar radiation and air temperature data obtained from MERAA-2 were used as an input. . .", Please explain these data are half hour or hourly data.

3.1 "BVOC mixing ratios and flux", . . .and fluxes?

3.3 Comparison of measured BVOC fluxes with MEGAN2.1, the emission unit is mg compound m−2 h−1 for isoprene and total monoterpenes. It is mg C m−2 h−1? If not, it is suggested to use the unified unit in the full text.

Lines 251-253, "measured isoprene and monoterpenes emission activity factors showed slightly different pattern during midday (isoprene is relatively constant from 11:00 to 14:00", please explain why the emission activity factor of isoprene is relatively constant, while monoterpenes is a peak around 11:00 to 12:00).

---

## Referee Comment (RC2) · Anonymous Referee #2 · 28 Mar 2020

This manuscript presents the flux measurements of biogenic VOCs in a tropical forest in eastern Amazonia. It reports isoprene and total monoterpene fluxes measured by a PTR-ToF-MS using the eddy covariance method. The data set, even though collected for two weeks only, is an important for understanding BVOC emissions for a highly active emission area with a substantial uncertainty on isoprene and monoterpene emissions. Thus, the data itself would be worth publishing, as it'd provide one more data point for the flux measurements, emission factors for two important biogenic compounds for a critical site. The paper focuses on the basic analysis by reporting mixing ratios and fluxes of two important BVOCs, and comparing the flux measurements with the MEGAN emission model prediction. Beyond reporting data in the rarely

observed station, the paper suggests the importance of the MEGAN2.1 1-km emission factor product and the large errors in the MEGAN2.1 PFT emission scheme, in terms of when compared to field measurements in local to regional scale studies. In summary, I'd recommend publishing the manuscript if authors could address the following concerns:

1. Measurement uncertainties (concentrations and fluxes) need to be better documented. This is important, particularly when interpreting the results comparing to the model prediction.

2. How was total monoterpene quantified if only one pinene was calibrated? What kind of assumptions were made when reporting the total monoterpene, and what is the associated uncertainty?

3. It looks like the MEGAN prediction is driven by the reanalysis meteorological field for air temperature and solar radiation data. Did the observations use the reanalysis meteorological field too? How much of the discrepancy between model and observation is actually from the difference in temperature and light data as the input data? Comparison in those process levels could really help improve our understanding in BVOC emissions, rather than simply suggesting models are overpredicting or underpredicting.

4. Several places in the manuscript mentioned 'Amazonian tree species' and their variation and distribution, and the site-specific emission factors appear to be the conclusion this manuscript tries to highlight. Can the authors give more descriptions on the tree types for the measurement site? The current description is very generic.

5. One important conclusion is that it seems like the 1-km resolution emission factor product is better than the 16 PFT emission factor method, which is not too surprising. Can the authors comment on how the 1-km product was derived in the first place for this region? Perhaps this could help shed light on how to estimate EF for those places without direct flux measurements?

Other specific comments:

a) The paper refers to MERAA-2 as the reanalysis meteorological data, but it should actually be MERRA-2 as its 'official' name.

b) Section 2.3 equation (2). The air density is needed for the eddy covariance calculation. How did the air density come from here? Was it measured or estimated?

c) Similar to Question 1. Need to describe the system errors involved in the EC flux error estimates, and discuss how the measurement errors affect model: observation comparison and other conclusions.

d) Section 3.1 title: consider 'BVOC mixing ratios and fluxes'?

e) Figure 4. Again similar to Question 3 above. Is the gamma temperature (and light) the same between measurements and models here? Are the difference driven by MEGAN input data such as light and temperature or by other processes?

---

## Author Comment (AC2) · 20 May 2020

PTR-TOF-MS eddy covariance measurements of isoprene and monoterpene fluxes from an Eastern Amazonian rainforest

Chinmoy Sarkar et al. (acp-2019-1161)

We would like to thank the Editor (Dr. Holzinger) and both the referees for appreciating and highlighting the importance of our work and for recommending the manuscript for publication in ACP subject to minor revisions. We have found several comments and suggestions from both the referees very helpful and these are now reflected in the revised submission. The comments from the referees are in black and our responses are in blue. The texts that have been added to the manuscript are in **bold blue**.

**Editor (Dr. Holzinger):**

Provide sensitivities of isoprene and monoterpenes in cps/ppb also.

Sensitivities for isoprene and monoterpenes are $12.8 \pm 0.32$ ncps/ppb **(54.9 $\pm$ 1.32 cps/ppb)** and $14.7 \pm 0.86$ ncps/ppb **(63.1 $\pm$ 3.69 cps/ppb)**, respectively. This information is now added in lines 95-96 of the revised manuscript.

**Referee 1:**

This study reports BVOC concentrations and fluxes measured at an eastern Amazon rainforest in the Tapajós National Forest during wet to dry transition period and compare the observations with the BVOC fluxes estimated with the MEGAN 2.1 model. The comparison between the measured and the modeled isoprene and monoterpene fluxes are discussed and the importance of site-specific parameters for BVOC flux estimation is highlighted. In addition, the measured BVOC fluxes were compared with previously reported measurements from the Amazon rainforest. The results demonstrate the importance of site-specific vegetation emission factors for accurately simulating BVOC fluxes in regional and global BVOC emission models. In general, all measured BVOC concentrations and emission fluxes are very valuable for accurate understanding of BVOCs and their chemistry and photochemistry, BVOC model simulations at regional and global scales.

Thank you for highlighting and appreciating the importance of our work.

Some minor comments and suggestions are as follows.

Line 100, "The total uncertainty for isoprene and monoterpenes were estimated to be < 20% for this study", Please explain what is the total uncertainty for isoprene and monoterpenes, including the measurements, e.g., sample collection, pump, etc.?

This information is already provided in the previous sentence (lines 100-102 of the original submission and lines 103-105 of the revised manuscript) which reads as follows:

"The total uncertainty for the calibrated VOCs were estimated by considering the accuracy error of the VOC standard, the instrumental precision error and the flow fluctuations of the mass flow controllers (MFCs) during the calibrations."

Line 141, what is the EC flux error estimation? Please explain it.

A detailed description of the EC flux error estimation is already provided in Park et al. (2013) and is mentioned in lines 140-143 of the original submission:

"As per the procedure described in Park et al. (2013), EC flux error estimations were carried out by considering the systematic errors due to inlet dampening, sensor separation, instrument response time, random noise in EC flux and uncertainties associated with concentration determinations."

We have now added the information for individual errors for this dataset in the revised manuscript and the revised sentence (lines 143-146 of the revised manuscript) reads as follows:

"As per the procedure described in Park et al. (2013), EC flux error estimations were carried out by considering the systematic errors due to inlet dampening **(< 12%)**, sensor separation **(< 1.1%)**, instrument response time **(0.3%)**, random noise **(isoprene < 4.0% ; monoterpenes < 0.2 %)** in EC flux and uncertainties associated with concentration determinations."

Line151, "Solar radiation and air temperature data obtained from MERAA-2 were used as an input…", Please explain these data are half hour or hourly data.

Hourly data were used as an input to run MEGAN2.1. We have now mentioned this in line 155 of the revised manuscript:

"Solar radiation and air temperature **hourly** data obtained from **MERRA-2** were used as an input to run MEGAN 2.1 for this study."

3.1 "BVOC mixing ratios and flux", …and fluxes?

We have now revised the title of section 3.1. It reads as "BVOC mixing ratios and **fluxes**" in the revised version.

3.3 Comparison of measured BVOC fluxes with MEGAN2.1, the emission unit is mg compound m-2 h-1 for isoprene and total monoterpenes. It is mg C m-2 h-1? If not, it is suggested to use the unified unit in the full text.

Done. We have now used unified unit (mg C m$^{-2}$ h$^{-1}$) in the full text.

Lines 251-253, "measured isoprene and monoterpenes emission activity factors showed slightly different pattern during midday (isoprene is relatively constant from 11:00 to 14:00", please explain why the emission activity factor of isoprene is relatively constant, while monoterpenes is a peak around 11:00 to 12:00).

The reason for the slightly different pattern of the measured isoprene and monoterpene emission activity factors during midday is unknown. Long-term BVOC flux measurements at this site in the future will provide more insights towards explaining these slight discrepancies in the diel profiles of emission activity factors.

**Referee 2:**

This manuscript presents the flux measurements of biogenic VOCs in a tropical forest in eastern Amazonia. It reports isoprene and total monoterpene fluxes measured by a PTR-ToF-MS using the eddy covariance method. The dataset even though collected for two weeks only, is an important for understanding BVOC emissions for a highly active emission area with a substantial uncertainty on isoprene and monoterpene emissions. Thus, the data itself would be worth publishing, as it'd provide one more data point for the flux measurements, emission factors for two important biogenic compounds for a critical site. The paper focuses on the basic analysis by reporting mixing ratios and fluxes of two important BVOCs and comparing the flux measurements with the MEGAN emission model prediction. Beyond reporting data in the rarely observed station, the paper suggests the importance of the MEGAN2.1 1-km emission factor product and the large errors in the MEGAN2.1 PFT emission scheme, in terms of when compared to field measurements in local to regional scale studies. In summary, I'd recommend publishing the manuscript if authors could address the following concerns:

We thank the referee for appreciating and highlighting the importance of our work.

1. Measurement uncertainties (concentrations and fluxes) need to be better documented. This is important, particularly when interpreting the results comparing to the model prediction.

The measurement uncertainties for concentrations are already mentioned in lines 100-103 of the original submission (lines 103-106 of the revised manuscript) and reads as follows:

"The total uncertainty for the calibrated VOCs were estimated by considering the accuracy error of the VOC standard, the instrumental precision error and the flow fluctuations of the mass flow controllers (MFCs) during the calibrations. The total uncertainty for isoprene and monoterpenes were estimated to be $< 20\%$ for this study."

The measurement uncertainties for fluxes are now mentioned in lines 143-146 of the revised manuscript:

"As per the procedure described in Park et al. (2013), EC flux error estimations were carried out by considering the systematic errors due to inlet dampening **($< 12\%$)**, sensor separation **($< 1.1\%$)**, instrument response time **(0.3%)**, random noise **(isoprene: $< 4.0\%$ ; monoterpenes $< 0.2$ %)** in EC flux and uncertainties associated with concentration determinations."

2. How was total monoterpene quantified if only one pinene was calibrated? What kind of assumptions were made when reporting the total monoterpene, and what is the associated uncertainty?

The estimation of total monoterpenes was made by considering the fragmentation at $m/z = 137.132$ ($\sim 40\%$) and 81.070 ($\sim 60\%$). This information is already mentioned in detail in lines 93-97 of the original submission (lines 96-100 of the revised manuscript):

"Sensitivity for total monoterpenes were estimated by considering the fragmentation at $m/z = 137.132$ and 81.070. Therefore, the signal measured at $m/z = 137.132$ was scaled by 2.5, as the calibrations at the above-mentioned instrumental settings showed $\sim 40\%$ of total monoterpenes

were detected at m/z = 137.132. This fragmentation pattern of monoterpenes is in agreement with previously reported studies with similar operating conditions (Tani et al., 2004; Sarkar et al., 2016)."

The measurement uncertainties for concentrations are already mentioned in line 100-103 of the original submission (lines 103-106 of the revised manuscript) and reads as follows:

"The total uncertainty for the calibrated VOCs were estimated by considering the accuracy error of the VOC standard, the instrumental precision error and the flow fluctuations of the mass flow controllers (MFCs) during the calibrations. The total uncertainty for isoprene and monoterpenes were estimated to be < 20% for this study."

3. It looks like the MEGAN prediction is driven by the reanalysis meteorological field for air temperature and solar radiation data. Did the observations use the reanalysis meteorological field too? How much of the discrepancy between model and observation is actually from the difference in temperature and light data as the input data? Comparison in those process levels could really help improve our understanding in BVOC emissions, rather than simply suggesting models are overpredicting or underpredicting.

Since we did not have measured solar radiation data at the measurement site, hourly solar radiation data obtained from the MERRA-2 satellite were used as an input in MEGAN2.1. We also compared the measured temperature and the temperature data obtained from MERRA-2 and found a difference of ~ 15% between them. If we assume that this 15% difference is constant in the solar radiation data too, then we can use this approximation to the solar radiation data obtained from MERRA-2 to roughly estimate solar radiation at the measurement site. When we used this estimated solar radiation data as an input in MEGAN2.1, we found an overestimation of ~ 10% to the isoprene and total monoterpene fluxes compared to the fluxes we have reported in the manuscript.

4. Several places in the manuscript mentioned 'Amazonian tree species' and their variation and distribution, and the site-specific emission factors appear to be the conclusion this manuscript tries to highlight. Can the authors give more descriptions on the tree types for the measurement site? The current description is very generic.

Among the 151 group of specimens found in the measurement area, the dominant tree species were *Erisma uncinatum Warm.*, *Carapa guianensis Aubl.*, *Manilkara huberi (Ducke) A. Chev.*, *Protium sp.*, *Lecythis lurida (Miers) S. A. Mori*, *Tachigali spp.*, *Ocotea sp.*, *Parkia sp.*, *Couratari sp.* and *Astronium gracile Engl.*. A detail description of different tree species in the Forest Management unit, with the respective number of trees (N), absolute density (DA), relative density (DR), absolute dominance (DoA), relative dominance (DoR), absolute frequency (FA), relative frequency (FR) and importance value index relative (IVI%), in alphabetical order of families and in decreasing order of IVI% can be obtained from a previous work by Gonçalves and Santos (2008).

We have now added the following sentence in the revised manuscript (lines 71-74):

**"Among the 151 group of specimens found in the measurement area, the dominant tree species were *Erisma uncinatum Warm.*, *Carapa guianensis Aubl.*, *Manilkara huberi (Ducke) A. Chev.*, *Protium sp.*, *Lecythis lurida (Miers) S. A. Mori*, *Tachigali spp.*, *Ocotea sp.*, *Parkia sp.*, *Couratari sp.* and *Astronium gracile Engl.* (Gonçalves and Santos, 2008)."**

**Gonçalves, Fábio Guimarães, & Santos, João Roberto dos. (2008). Floristic composition and structure of a sustainable forest management unit in the Tapajós National Forest, Pará. Acta Amazonica, 38 (2), 229-244. https://doi.org/10.1590/S0044-59672008000200006.**

5. One important conclusion is that it seems like the 1-km resolution emission factor product is better than the 16 PFT emission factor method, which is not too surprising. Can the authors comment on how the 1-km product was derived in the first place for this region? Perhaps this could help shed light on how to estimate EF for those places without direct flux measurements?

The MEGAN2.1 1km resolution global emission factor map (Guenther et al., 2012) was generated by integrating emissions data reported in the literature for each of the 862 global ecoregions in the scheme developed by Olson et al. (2001). The BVOC emissions data that Guenther et al. (2012) used to characterize the "Tapajós-Xingu moist forests" eco-region, which contains the Tapajós flux tower field site, includes measurements reported by Greenberg et al. (2004) and Harley et al. (2004).

This information is now added in lines 242-246 of the revised manuscript:

**"The MEGAN2.1 1km resolution global emission factor map (Guenther et al., 2012) was generated by integrating emissions data reported in the literature for each of the 862 global ecoregions in the scheme developed by Olson et al. (2001). The BVOC emissions data that Guenther et al. (2012) used to characterize the "Tapajós-Xingu moist forests" ecoregion, which contains the Tapajós flux tower field site, includes measurements reported by Greenberg et al. (2004) and Harley et al. (2004)."**

**Olson, D. M., Dinerstein, E., Wikramanayake, E. D., Burgess, N. D., Powell, G. V. N., Underwood, E. C., D'Amico, J. A., Itoua, I., Strand, H. E., Morrison, J. C., Loucks, C. J., Allnutt, T. F., Ricketts, T. H., Kura, Y., Lamoreux, J. F., Wettengel, W. W., Hedao, P., and Kassem, K. R.: Terrestrial ecoregions of the world: a new map of life on Earth. Bioscience 51(11):933-938, 2001.**

Other specific comments:

a) The paper refers to MERAA-2 as the reanalysis meteorological data, but it should actually be MERRA-2 as its 'official' name.

Thanks a lot for pointing it out. We have now corrected it and mentioned **MERRA-2** throughout the revised manuscript and in Figure 2.

b) Section 2.3 equation (2). The air density is needed for the eddy covariance calculation. How did the air density come from here? Was it measured or estimated?

Air density was estimated based on the ideal gas law (PV = nRT) using the measured air temperature (T) and pressure (P) at the site.

c) Similar to Question 1. Need to describe the system errors involved in the EC flux error estimates and discuss how the measurement errors affect model: observation comparison and other conclusions.

System errors involved in the EC flux error estimation is already described while replying to Question 1 and reads as follows (lines 143-146 of the revised manuscript):

"As per the procedure described in Park et al. (2013), EC flux error estimations were carried out by considering the systematic errors due to inlet dampening **(< 12%)**, sensor separation **(< 1.1%)**, instrument response time **(0.3%)**, random noise **(isoprene: < 4.0% ; monoterpenes < 0.2 %)** in EC flux and uncertainties associated with concentration determinations."

As discussed above, the total measured EC flux uncertainty for isoprene and monoterpenes were < 20%. The modeled uncertainty for isoprene was < 20% during daytime (08:00 – 16:00 LT) and > 20% during early morning (07:00 – 08:00 LT) and late afternoon/evening (16:00 – 18:00 LT). Similarly, the modeled uncertainty for monoterpenes was < 20% during daytime (08:00 – 15:00 LT) and > 20% during early morning (07:00 – 08:00 LT) and late afternoon/evening (15:00 – 18:00 LT).

d) Section 3.1 title: consider 'BVOC mixing ratios and fluxes'?

We have now revised the title of section 3.1. It reads as "BVOC mixing ratios and **fluxes**" in the revised version.

e) Figure 4. Again similar to Question 3 above. Is the gamma temperature (and light) the same between measurements and models here? Are the difference driven by MEGAN input data such as light and temperature or by other processes?

As discussed above while replying to Question 3, we did not have measured solar radiation data at the measurement site and therefore, used hourly solar radiation data obtained from the MERRA-2 satellite as an input in MEGAN2.1 and the diel profiles of estimated isoprene and monoterpene emission activity factors are shown in Figure 4. We also compared the measured temperature and the temperature data obtained from MERRA-2 and found a difference of ~ 15% between them. If we assume that this 15% difference is constant in the solar radiation data too, then we can use this approximation to the solar radiation data obtained from MERRA-2 to roughly estimate solar radiation at the measurement site. When we used this estimated solar radiation data as an input in MEGAN2.1, we found an overestimation of ~ 10% to the isoprene and monoterpene fluxes compared to the fluxes we have reported in the manuscript.